# Direct observation of ultrafast singlet exciton fission in three dimensions

Arjun Ashoka[1], Nicolas Gauriot[1], Aswathy V. Girija [1], Nipun Sawhney[1], Alexander J. Sneyd[1], Kenji Watanabe [2], Takashi Taniguchi [3], Jooyoung Sung [4], Christoph Schnedermann [1] ✉ & Akshay Rao [1] ✉

We present quantitative ultrafast interferometric pump-probe microscopy capable of tracking of photoexcitations with sub-10 nm spatial precision in three dimensions with 15 fs temporal resolution, through retrieval of the full transient photoinduced complex refractive index. We use this methodology to study the spatiotemporal dynamics of the quantum coherent photophysical process of ultrafast singlet exciton fission. Measurements on microcrystalline pentacene films grown on glass ($SiO_2$) and boron nitride (hBN) reveal a 25 nm, 70 fs expansion of the joint-density-of-states along the crystal $a,c$-axes accompanied by a 6 nm, 115 fs change in the exciton density along the crystal $b$-axis. We propose that photogenerated singlet excitons expand along the direction of maximal orbital $\pi$-overlap in the crystal $a,c$-plane to form correlated triplet pairs, which subsequently electronically decouples into free triplets along the crystal $b$-axis due to molecular sliding motion of neighbouring pentacene molecules. Our methodology lays the foundation for the study of three dimensional transport on ultrafast timescales.

Elucidating the three-dimensional transport of excitations in condensed matter is key to advancements in our understanding and utilisation of functional materials ranging from novel quantum systems to next-generation optoelectronic materials[1–4]. Of particular interest are quantum coherent processes in heterogeneous, disordered systems which require both ultrafast time resolution and local measurements to study and understand their transport characteristics[5–7]. Singlet exciton fission is a widely studied example of such a process that has gained relevance in the fields of photovoltaics and quantum computing. Here a photogenerated singlet exciton converts to an electronically and spin entangled pair of triplets at nearly half the singlet energy on ultrafast timescales. The correlated triplet pair then separates into individual triplet excitons through the loss of electronic and spin coherence[8–11]. Numerous ultrafast studies have probed the coherent dynamics of the singlet fission process, however direct observation of the real space three-dimensional spatial dynamics of ultrafast singlet fission remains an outstanding goal[8,10].

The field of optical pump-probe microscopy, which extends standard pump-probe spectroscopy to a microscope geometry, is well suited to study coherent transport processes in condensed matter systems, as it provides direct visualisation of the transport of excitations[12–14]. Conventionally, optical pump-probe microscopy has used point-scanning methodologies which provide a two-dimensional picture of the transport process with 10 s of nm resolution and 100 fs time resolution. In contrast, reports of widefield optical pump-probe microscopy have demonstrated that interferometric contrast could provide a way to visualise three-dimensional transport[14]. However, while the use of phenomenological Gaussian point-spread functions to describe the measured images can yield sub-10 nm lateral precision, it cannot quantify out-of-plane transport[12,14–17]. Moreover, the measured images naturally arise from changes in the full complex refractive

[1]Cavendish Laboratory, University of Cambridge, J. J. Thomson Avenue, Cambridge CB3 0HE, UK. [2]Research Center for Functional Materials, National Institute for Materials Science, 1-1 Namiki, Tsukuba 305-0044, Japan. [3]International Center for Materials Nanoarchitectonics, National Institute for Materials Science, 1-1 Namiki, Tsukuba 305-0044, Japan. [4]Department of Emerging Materials Science, DGIST, Daegu 42988, Republic of Korea. ✉e-mail: cs2002@cam.ac.uk; ar525@cam.ac.uk

index, $n + ik$, which cannot be retrieved using such phenomenological Gaussian point-spread-functions, which in turn rules out the quantitative measurement of the transient joint-density-of-states (JDOS).

Here we introduce quantitative interferometric ultrafast pump-probe microscopy. Using a first-principles analytic optical model, we show it is possible to fully describe these interferometric pump-probe images and quantify changes in the full transient complex refractive index $n(t) + ik(t)$ and retrieve both lateral and out-of-plane transport with sub-10 nm precision, with 15 fs time resolution (see Supplementary Note 6 for localisation precision and Supplementary Note 4 for demonstration of time resolution). We report a full three-dimensional picture of ultrafast singlet exciton fission in polycrystalline pentacene films, suggesting that the photoexcited singlet exciton expands along the direction of maximal orbital $\pi$-overlap ($a,c$-axes) to form correlated triplet pairs, which subsequently electronically decouples into free triplets along the crystal $b$-axes due to lattice modes that drive the intermolecular sliding motion of neighbouring pentacene molecules.

## Results and discussion
### Experimental setup and optical model

Our experimental setup (Fig. 1a) is based on a transmission widefield pump-probe microscope equipped with an oil-immersion objective (numerical aperture = 1.1). A pump pulse (560 nm, 13 fs) is focused onto the sample with the objective to produce a near-diffraction-limited local photoexcitation (Supplementary Note 1). After a variable time delay, a counter-propagating widefield probe pulse (750 nm, 7 fs, ~20 μm at full-width-half-maximum) is transmitted through the sample and imaged onto an emCCD detector. The effect of the pump pulse is to photoinduce a three-dimensional spatially varying, local complex refractive index change, $\Delta \tilde{n} = \Delta n + i\Delta k$ (Fig. 1a). This index change weakly perturbs the time-delayed plane-wave probe pulse incident on the sample, leading to local changes in its phase and amplitude[18]. The large background unperturbed probe field interferes with this perturbed probe field to form a spatial interference pattern along the

propagation direction. The objective and imaging lens then image this spatial interference pattern combined with the attenuation of the probe on a camera.

Our optical model is based on the treatment of the diffraction-limited optical perturbation produced by the pump pulse in a thin film semiconductor as a well-defined local complex refractive index perturbation, similar to a gold nanoparticle or polystyrene bead (Fig. 1a). Before the pump pulse arrives, when the system is in the ground state (pump-off), the polarisation $P$ measured by the probe is given by, $P = \epsilon_0 \chi^{(1)} E$. Using $D = \epsilon E = \epsilon_0 E + P$, the ground state dielectric function $\epsilon_{\text{off}}$ is therefore given by, $\epsilon_{\text{off}} = \epsilon_0 (1 + \chi^{(1)})$. After the arrival of the pump pulse, the system is in the excited state (pump-on), the polarisation $P$ measured by the probe is given by $P = \epsilon_0(\chi^{(1)} E + \chi^{(3)} E_{\text{pu}} E_{\text{pu}} E)$, where $E_{\text{pu}}$ is the pump electric field. Similarly, the excited state dielectric function $\epsilon_{\text{on}}$ is therefore given by, $\epsilon_{\text{on}} = \epsilon_0(1 + \chi^{(1)} + \chi^{(3)} E_{\text{pu}} E_{\text{pu}})$. As the probe pulse is always temporally separated from the pump, time-ordering allows us to treat the pump-on and pump-off probe signals as measures of the photoexcited and ground state dielectric functions (or complex refractive index) of the material, respectively. The overall perturbation to the dielectric function can therefore be written as,

$$\epsilon_{on} - \epsilon_{\text{off}} = \Delta\epsilon = \epsilon_0 \chi^{(3)} E_{\text{pu}}^2, \tag{1}$$

where $\chi^{(3)}$ is the third order non-linear susceptibility, $\epsilon_0$ is the permittivity of free space and $E_{\text{pu}}$ is the pump electric field[19,20]. In order to link this to the refractive index, we use the fact that as $\tilde{n} = \sqrt{\epsilon}$, $\Delta \tilde{n} = \frac{1}{2\sqrt{\epsilon}} \Delta\epsilon$ for small perturbations. As the time-integrated intensity of the pump absorbed by the material is given by $I_{\text{pu}} = \frac{c n_0 \epsilon_0}{2} E_{\text{pu}}^2$, the photoinduced refractive index change can be written as,

$$\Delta\tilde{n}(r,z) = \frac{1}{2n_0} \chi^{(3)} \frac{2}{c n_0} I_{\text{pu}}(r,z) \tag{2}$$

which importantly shows that upon photoexcitation, $\Delta \tilde{n}$ is proportional to the intensity of the pump pulse.

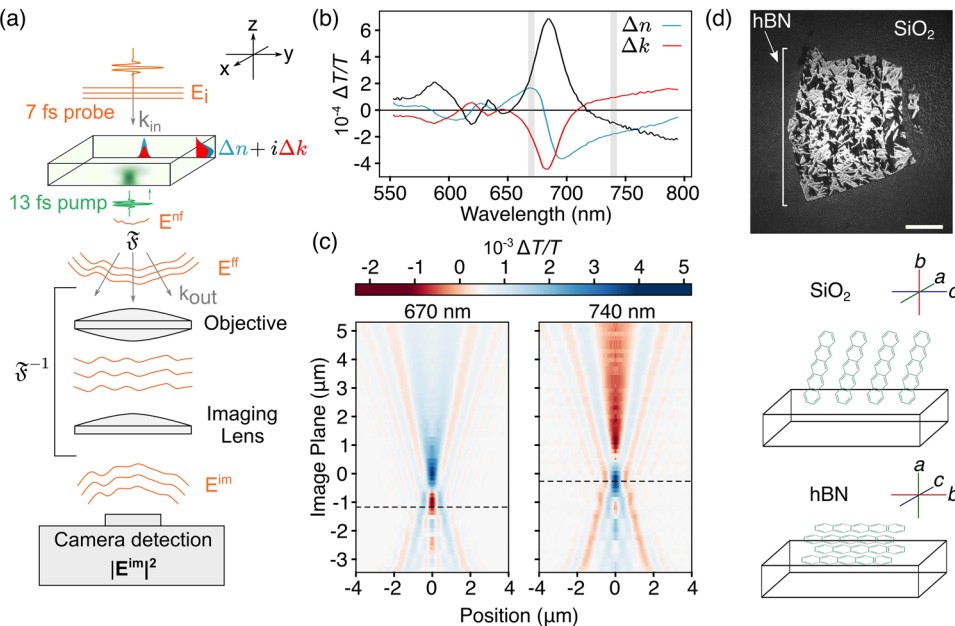

**Fig. 1 | Experimental setup and optical model. a** A schematic of the propagation of a time-delayed weakly interacting probe through a locally perturbed three-dimensional sample imaged on a camera. **b** Transient transmission and extracted transient complex refractive index spectra, measured bands indicated in grey. **c** Radially averaged differential transmission images measured at different planes through a film of pentacene 150 fs after photoexcitation. The radial averages are mirrored about the $x = 0$ plane for clarity. Dashed lines indicate the interferometrically enhanced planes chosen for measurement. **d** Microscope image of the microcrystalline domains of the measured pentacene film on hBN and SiO$_2$ substrates, with labelled crystal geometries relative to the substrate. (Scale bar = 20 μm).

As the intensity at the back-focal-plane of the objective is a $TEM_{00}$ mode, exploiting radial symmetry to average over polarisation effects, the pump is a focused Gaussian beam attenuated through the depth of the sample,

$$I_{pu}(r,z) = I_0 \left[ \frac{\sigma_0}{\sigma(z)} \right]^2 \exp\left[ \frac{-r^2}{2\sigma(z)^2} - \frac{\omega_{pu}\alpha z}{c} \right] \quad (3)$$

where $\sigma(z) = \sigma_0 \sqrt{1 + \left(\frac{z}{z_R}\right)^2}$, $z_R$ being the Rayleigh range, $\omega_{pu}$ is the pump frequency and $\alpha$ is dimensionless, thickness-corrected extinction coefficient for the pump. For thin film samples of thickness less than $z_R$, and ignoring in-plane anisotropy, $\sigma$ can be approximated to be constant $(\mathcal{O}(z/z_R)^2)$ through the thickness of the sample. We can therefore approximate the photoinduced refractive index change as,

$$\Delta\tilde{n}(r) = (\Delta n_0 + i\Delta k_0) \exp\left[ \frac{-r^2}{2\sigma_0^2} - \frac{\omega_{pu}\alpha z}{c} \right] \quad (4)$$

over the sample length, where we have absorbed the constants and $\chi^{(3)}$ into the transient optical constants $\Delta n_0 + i\Delta k_0 = \frac{1}{2n_0}\chi^{(3)}\frac{2}{cn_0}I_0$. We compute the near-field electric field by multiplying the incident field with the laterally varying Fresnel complex transmission coefficients of a Gaussian disc situated about the photoexcitation's principal plane $z_c$,

$$E^{nf}(r,z_c) = E_i[(1 - r_1(r))[1 - r_2(r))]t(r), \quad (5)$$

where $t = e^{iL\tilde{n}\kappa}$ captures the attenuation of $E_i$ through the sample, $r_1 = \frac{n_{air} - \tilde{n}}{n_{air} + \tilde{n}}$ captures the reflection at the air-sample interface and $r_2 = \frac{\tilde{n} - n_s}{\tilde{n} + n_s}$, captures the substrate-sample interface. As the microscope system we study is an oil-immersion system, the substrate-oil interface just before the objective is near index matched and not taken into consideration. Here $n_s$ is the substrate refractive index, $L$ is the sample thickness and $\kappa$ is the probe wavevector. The amplitude and phase of $E^{nf}$, therefore, encodes three-dimensional spatial information of photoexcited carrier transport through $\Delta\tilde{n}(r)$ and $z_c$.

The probe electric field at the objective's input aperture can be approximated as the Fourier transform of the near-field electric field, $E^{ff}(k_r,z_c) = \mathfrak{F}(E^{nf}(r,z_c))$, where $k_r$ is the spatial frequency of the probe in the radial direction and $\mathfrak{F}$ is the Fourier transform. To study the spatial interference of the probe in different planes in the object space where the field is interferometrically enhanced, we calculate the far-field electric field of the plane $\Delta z = z_0 - z_c$ by propagating the plane-wave decomposition an extra distance $\Delta z$, $E^{ff}(k_r,\Delta z) = e^{i\Delta z\sqrt{\kappa^2 - k_r^2}}E^{ff}(k_r,z_c)$.

To calculate the image on the camera by the objective-imaging lens system, we compute an inverse Fourier transform after filtering the high spatial frequencies as the objective's NA specifies, yielding,

$$E^{im}(r',\Delta z,n,k) = \int_{-\kappa NA}^{\kappa NA} E^{ff}(k_r,\Delta z)e^{-ik_r r'}dk_r \quad (6)$$

which is a simplified version of the Richards–Wolf integral[21–23].

We calculate the widefield normalised differential transmitted image of a photoexcited refractive index change (at a given probe wavelength), $\Delta\tilde{n}(r) = \Delta n(r) + i\Delta k(r)$ centred at $z_c = z_0 - \Delta z$ on a spatially constant, static background $\tilde{n}_0(r) = n_0 + ik_0$ centred about $z_0$,

$$\frac{\Delta T}{T}(r',z_0 - \Delta z,\Delta\tilde{n}) = \frac{|E^{im}(r',\Delta z,\tilde{n}_0 + \Delta\tilde{n}(r))|^2}{|E^{im}(r',z_0,\tilde{n}_0)|^2} - 1. \quad (7)$$

Experimental control over the image plane $z_0$ is achieved by translating the imaging lens in the infinity space of the objective to relay different conjugate planes to the camera[24]. The effective axial distance accessible by moving the imaging lens by a distance $z'$ is given by the axial magnification of the imaging system (Supplementary Note 7). Ensuring that neither the imaging lens nor the objective move during a pump-probe measurement fixes $z_0$, allowing us to track changes in $z_c$, $\sigma$ and transient refractive index $\Delta\tilde{n}$ as a function of pump-probe delay.

Our model, therefore, characterises the normalised differential transmitted images in terms of the physical parameters $\Delta\tilde{n}(r,\sigma,z_0)$, the probe wavelength and the sample thickness. We benchmark our model against Finite-Difference-Time-Domain (FDTD) calculations and find excellent agreement, demonstrating that our near-field approximations and the computational challenging aberrations are systematically cancelled in the differential far-field images (Supplementary Note 2 and 3). Critically, this enables us to extract the three-dimensional transport and $\Delta n_0 + i\Delta k_0$ through fitting the measured data set, which would be prohibitively computationally challenging using purely FDTD methods.

The process of ultrafast singlet fission is considered to occur in three steps. First, the photogenerated singlet $S_1$ converts to an electronically and spin entangled triplet pair state $TT$. Second, through the loss of electronic correlation the $TT$ state converts to a spatially separated triplet pair $T...T$. Finally, through the loss of spin correlation, the spatially separated spin entangled triplet pair $T...T$ separates into two uncorrelated triplets $T + T$[8]. As spin coherence is typically lost on longer timescales than the electronic coherence in polyacenes and as we probe the electronic states through their optical transitions, we solely focus on the loss of electronic correlation from $TT$ to $T...T$ and make no comment on the spin correlation[8]. We study the dynamics of singlet exciton fission in two well-explored spectral bands of the archetype organic semiconductor pentacene: the photobleaching band at 670 nm and the photoinduced absorption (PIA) band at 740 nm (Fig. 1b)[25]. It has been established that the 670 nm band primarily tracks the ground-state bleach and hence contains the JDOS of both the photoexcited singlet $S_1$ and resulting entangled $TT$ and uncorrelated $T...T$ triplet pairs[25,26]. The 740 nm feature, however, is not present immediately after photoexcitation and tracks the JDOS of only the entangled $TT$ and separated $T...T$ triplet pair through transitions to higher lying triplet states. We describe the transitions in these systems through their JDOS rather than as single energetic transitions as in extended thin film systems, the molecules are not isolated and the density of states cannot be treated as one-dimensional.

Analysis of the transient transmission spectra based on a Kramers−Kronig differential dielectric function predicts $\Delta n > 0$ and $\Delta k < 0$ at 670 nm and $\Delta n < 0$ and $\Delta k > 0$ at 740 nm (Fig. 1b)[27]. We demonstrate our interferometric sensitivity to the photoexcitation by measuring the differential transmitted images 150 fs after photoexcitation at both wavelengths through several image planes (Fig. 1c), demonstrating three dimensional point-spread-functions that are reminiscent of those reported in state-of-the-art static interferometric scattering microscopy[28]. We access different image planes by translating the imaging lens in the infinity space of our microscope to relay different planes from the image space onto the camera, similar to the concept of remote focusing[24]. The degree of interferometric contrast determines our ability to resolve out-of-plane carrier transport and accurately retrieve the transient refractive indices. We, therefore, study spectral bands with a non-zero real refractive index change and a suitable imaging plane within the axial focus (Fig. 1c, dashed lines) (Supplementary Note 7)[19,29].

In order to unravel the three-dimensional dynamics of singlet fission in microcrystalline pentacene films, we study pentacene films evaporated on $SiO_2$ and on hexagonal boron nitride (hBN). On $SiO_2$ pentacene crystallises with its $b$-axis near-perpendicular to the substrate, whereas on hBN pentacene crystallises with the $b$-axis in-plane (Fig. 1d and Supplementary Note 8)[30]. This enables us to validate any

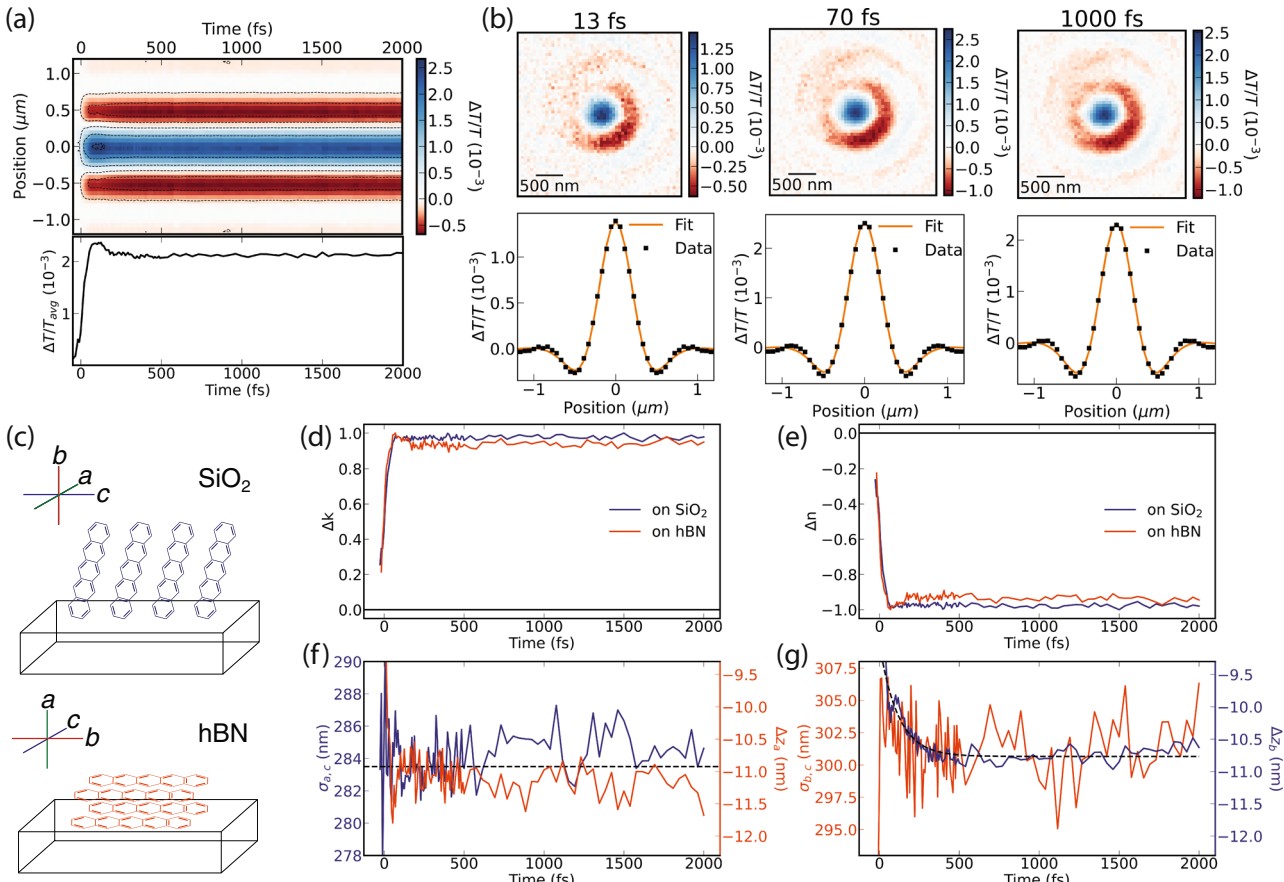

**Fig. 2 | Triplet decoupling dynamics at 740 nm and 250 μJ cm⁻². a** Measured radially averaged $\Delta T/T$ map and spatially averaged signal kinetic. **b** Fit of the optical model to the radial average at 13, 70, and 1000 fs. **c** Schematic of the measured pentacene film on hBN and SiO₂ substrates, with labelled crystal geometries relative to the substrate. **d, e** Transient optical constants display the expected signs based on Fig. 1b and show an ultrafast rise followed by a constant JDOS. **f** Transport in the

$a,c$-crystal plane measured in-plane on SiO₂ (blue) and along the $a$-crystal direction measured out-of-plane on hBN (orange) is absent. **g** Transport in the $b,c$-crystal plane measured in-plane on hBN (orange) and along the $b$-crystal direction measured out-of-plane on SiO₂ (blue) displays a few nm 115 fs change in the exciton density (dashed line fits an exponential).

measured out-of-plane transport against in-plane transport in the orthogonal orientation as well as confirm the direction of the out-of-plane transport. We begin by studying the uncongested PIA band at 740 nm to establish the spatial dynamics of the triplet excitons and use this information to study the more complicated dynamics of the ground state bleach band at 670 nm.

### Loss of triplet pair electronic correlation

We study the $TT$ and $T{\dots}T$ exciton dynamics in the 740 nm PIA band (on both SiO₂ and hBN) at 250 μJ cm⁻² where singlet exciton fission is the dominant photophysical process (Fig. 2)[31]. At these densities, there is on average one photoexcitation per 100 pentacene molecules. As shown in Fig. 2a, there are spatiotemporal changes to the signal during the first 500 fs. Our model is able to capture the measured differential transmission image in this regime (Fig. 2b) (see Supplementary Note 5 for details on the fitting procedure). A PIA band is expected to result from an increased JDOS, as the triplets are formed via the fission process and as the JDOS $\propto \mathrm{Im}(\tilde{\epsilon})$, where $\tilde{\epsilon}$ is the dielectric function and $\tilde{\epsilon} = \tilde{n}^2$, $\Delta k > 0$ is expected at 740 nm. We retrieve the expected signs of $\Delta n$ and $\Delta k$ (Fig. 1b) and find an ultrafast rise due to formation of triplets and a subsequently constant JDOS, identical between both studied crystal orientations as anticipated (Fig. 2c–e). Transport of the photoexcitation in the $a,c$-crystal plane measured through $\sigma_{a,c}$ on SiO₂ and $\Delta z_a$ on hBN (Fig. 2c, f) is absent to within our noise floor of 4 nm peak-to-peak. However, transport along the $b$-crystal direction measured through $\Delta z_b$ on SiO₂ displays

clear few nm transport with a timescale of 115 ± 11 fs (Fig. 2g). When measured in the rotated crystal orientation on hBN, this transport appears as the 6 nm change in the exciton density in the $b,c$-crystal plane measured through $\sigma_{b,c}$ (Fig. 2g).

The 6 nm, 115 fs change in the exciton density along the crystal $b$-axis at 740 nm must correspond to a photophysical process relating exclusively to the triplet population. This process must be distinct from the formation of the $TT$ state which would be correlated with the rise of this spectral feature. Further, the 115 fs timescale is distinct from previously reported $TT$ formation timescales in pentacene[25]. Hence this process must be related to the loss of electronic correlation in the triplet pair. Recent reports suggest that a 1 THz (1-ps period) lattice vibration in pentacene crystals associated with sliding motion of neighbouring pentacene molecules along the crystal $b$-axis modulates the $\pi$-overlap and therefore the J-coupling between adjacent pentacene molecules, resulting in triplet pair separation from $TT \rightarrow T{\dots}T$[32]. Our measured 6 nm change in the spatial triplet exciton density along the same crystal $b$-axis over 500 fs could be related to a lattice distortion arising from this 1 THz mode which changes the local excitonic density along this axis. A full 1-ps oscillation period of the 1 THz sliding mode is not needed to decouple the $TT$ state to the $T{\dots}T$ state, and as we show below, free triplets are formed within 200 fs of photoexcitation. We additionally note that the triplet exciton in pentacene is known to be polarised along the $b$-axis which further suggests a strong change in polarisability due to this mode[33]. Further theoretical investigations of the response of the excitonic wavefunction to phonon

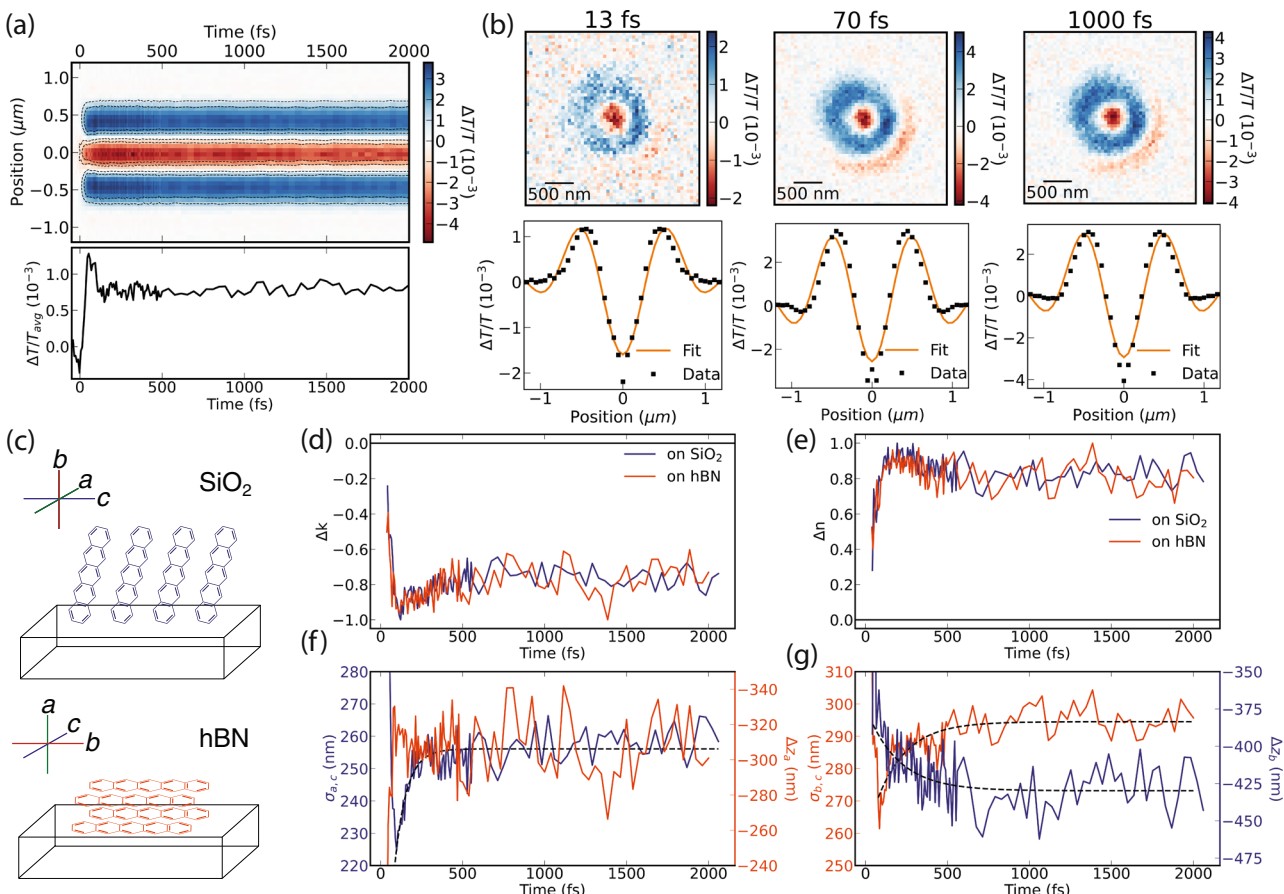

**Fig. 3 | Singlet fission dynamics at 670 nm and 250 μJ cm⁻². a** Measured radially averaged $\Delta T/T$ map and spatially averaged signal kinetic. **b** Fit of the optical model to the radial average at 13, 70 and 1000 fs. **c** Schematic of the measured pentacene film on hBN and SiO₂ substrates, with labelled crystal geometries relative to the substrate. **d, e** Transient optical constants display the expected signs based on Fig. 1b along with a sub-1-ps feature associated with the singlet exciton fission process and subsequent constant JDOS. **f** Transport in the $a,c$-crystal plane measured in-plane on SiO₂ (blue) and along the $a$-crystal direction measured out-of-plane on hBN (orange) displays a 25 nm 70 fs expansion, which can be resolved in $\sigma_{a,c}$ (dashed line is exponential fit). **g** Transport in the $b,c$-crystal plane measured in-plane on hBN (orange) and along the $b$-crystal direction measured out-of-plane on SiO₂ (blue) displays a 30 nm 180 fs expansion (dashed lines are exponential fit).

modes along this axis in the vibronic and transient delocalisation framework are called for[34–36].

## Singlet to correlated triplet transition

We study the spatiotemporal dynamics of the $S_1$ exciton via the 670 nm photobleaching band in the same fluence regime of 250 μJ cm⁻² (Fig. 3). As shown in Fig. 3a, there are spatiotemporal changes to the signal during the first 500 fs. Our model captures the measured differential transmission image and we are able to retrieve the correct signs of the $\Delta \bar{n}$ based on Fig. 1b for both crystal orientations (Fig. 3b–e)[27]. We recall that the 670 nm band is congested due to overlapping spectral features of the $S_1$, $TT$, $T...T$ and different oscillator strengths which makes a quantitative interpretation of $\Delta \bar{n}(t)$ at 670 nm challenging. Transport of the photoexcitations in the $a,c$-crystal plane of pentacene is tracked through $\sigma_{a,c}$ on SiO₂ and $\Delta z_a$ on hBN (Fig. 3c). We observe a 25 nm expansion in $\sigma_{a,c}$ with a timescale of $72 \pm 13$ fs, but no measurable correlated transport in $\Delta z_a$ (Fig. 3f). We note that the high localisation precision is significantly diminished on this spectral band as the required interferometric contrast can only be gained through substantially defocussing ($|\Delta z| > 500$ nm at 670 nm compared to $|\Delta z| <$ 50 nm at 740 nm), where the remote focussing model begins to fail (see Supplementary Note 7).

As the triplet ($TT$, $T...T$) excitons studied at 740 nm and display no transport in the $a,c$-crystal plane, any transport of in the $a,c$-crystal plane measured at the 670 nm ground state bleach necessarily arises from the $S_1$ exciton. The 70 fs timescale matches previously reported timescales of singlet fission that are sensitive to the $S_1 \rightarrow TT$ transition[10,25]. The presence of this feature in the crystal-$a,c$ plane is consistent with the fact that the J-coupling between pentacene molecules required for the $S_1 \rightarrow TT$ transition is maximal in direction of maximal π-overlap, i.e., the crystal-$a,c$ plane[37] (Fig. 1). This suggests that the $S_1 \rightarrow TT$ transition occurs in the crystal-$a,c$ plane which results in a spatial expansion of the exciton density potentially due to the formation of charge transfer (CT) states yielding a velocity $\mathcal{O}(10^5)$ m s⁻¹. Theoretical calculations of singlet exciton fission in solid pentacene using ab-inito Green's function methods predict a singlet exciton bandwidth of 100 meV over the unit cell, which yield an estimate of a group velocity of $\mathcal{O}(10^4)$ m s⁻¹, suggesting that this transport phenomena cannot be rationalised as typical coherent transport of the $S_1$ state[38]. The slower timescale of the expansion along the $b$-crystal direction measured in the $b,c$-crystal plane of pentacene is tracked through $\sigma_{b,c}$ on hBN is, however, difficult to interpret due to the overlapping transport of the triplets at 740 nm along the same direction (Fig. 3g).

## Triplet-triplet annihilation

Lastly, at high excitation densities, triplet-triplet annihilation (TTA) can dominate the photo-physics of pentacene and influence subsequent triplet transport and decay pathways. To study the effects of TTA in the ultrafast regime, we study the 740 nm band in the high

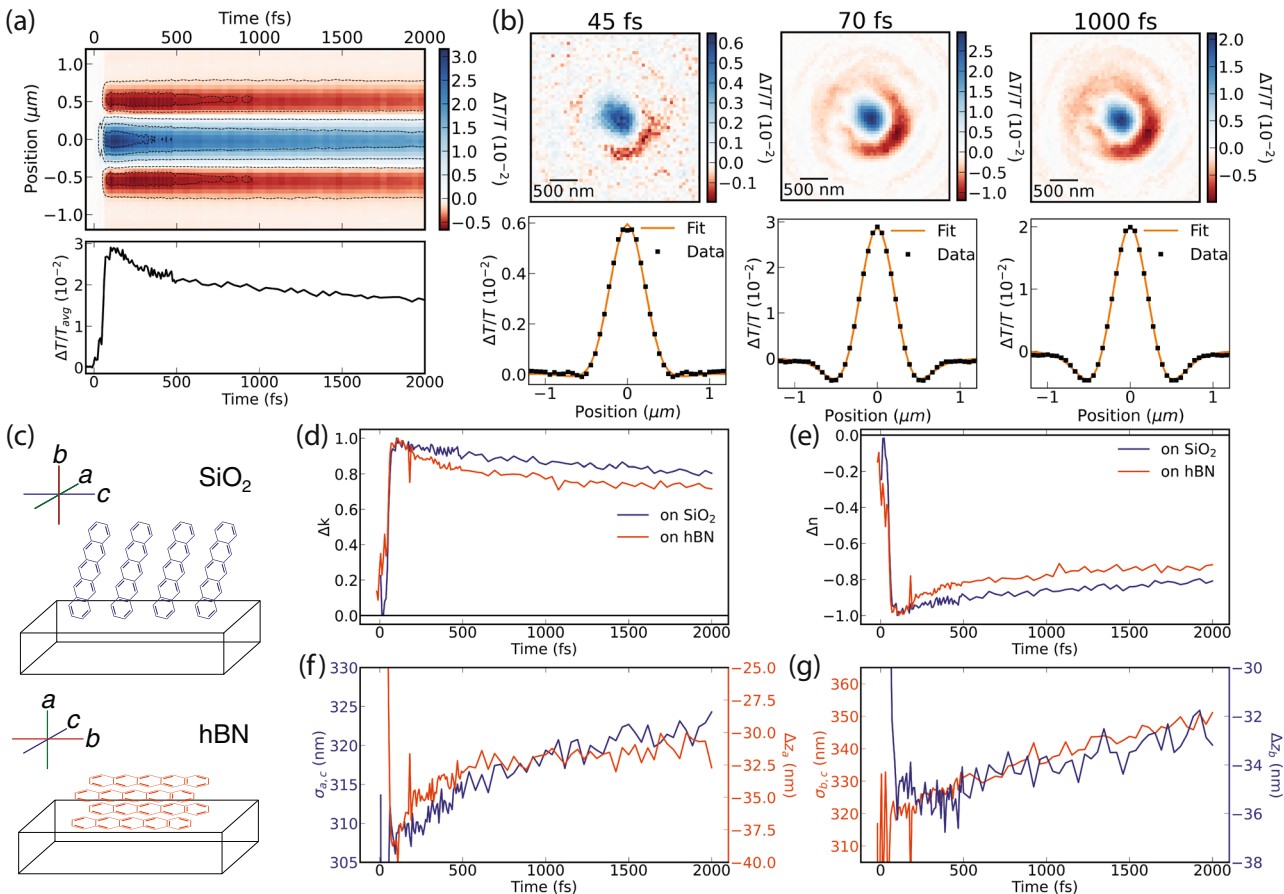

**Fig. 4 | Fast triplet annihilation dynamics at 740 nm and 750 μJ cm⁻².**
**a** Measured radially averaged $\Delta T/T$ map and spatially averaged signal kinetic. **b** Fit of the optical model to the radial average at 45, 70, and 1000 fs. **c** Schematic of the measured pentacene film on hBN and SiO₂ substrates, with labelled crystal geometries relative to the substrate. **d**, **e** Transient optical constants display the expected signs based on Fig. 1b and the JDOS decays due to triplet-triplet annihilation expected at these fluences. **f**, **g** Apparent transport due to the density-dependent TTA cross-section is seen on both SiO₂ and hBN and appears to be isotropic.

fluence regime at 750 μJ cm⁻² (Fig. 4)[31]. At these densities, there is on average one photoexcitation per 30 pentacene molecules and the spatiotemporal dynamics show a decay over the first picosecond (Fig. 4a). Even in the high-density limit, our model satisfactorily captures the measured data demonstrating robustness to the magnitude of the external $\bar{n}$ perturbation (Fig. 4b). We find that both $\Delta n$ and $\Delta k$ decay as a function of time as would be expected for a reduction in the JDOS of triplets through a TTA process, again, with near identical response irrespective of the crystal orientation (Fig. 4c–e).

As the exciton-exciton annihilation cross-section scales with the exciton density, the decay rate varies spatially over the photoexcitation profile. With time, this leads to a flat-topped exciton density profile, which when approximated as a Gaussian in Eqn. (4), results in an apparent expansion[39]. We resolve the apparent expansion from TTA in the $a,c$-crystal plane through both measured in-plane through $\sigma_{a,c}$ on SiO₂ and out-of-plane through $\Delta z_a$ on hBN (Fig. 4c, f). We resolve a similar expected apparent expansion in the $b$-crystal direction, suggesting that the TTA process is isotropic (Fig. 4g). The fast timescale for the onset of TTA suggests that such free triplets are formed within 200 fs in pentacene as TTA typically occurs for spatially separated triplets, i.e., $T...T$. This fast triplet annihilation process must be to higher lying triplet states and not back to the singlet manifold as the latter process is endothermic and would be expected to occur on longer timescales. The fast sub-200 fs transport along the $b$-crystal axis seen in the low-fluence regime at 250 μJ cm⁻² also appears to be

present in the high-density regime, suggesting that this feature is related to the singlet fission process.

To summarise, we have utilised interferometric pump-probe microscopy to reveal the three-dimensional picture of singlet exciton fission and triplet annihilation in microcrystalline pentacene films with sub-10 nm spatial precision and 15 fs temporal resolution. Our results suggest that the photoexcited singlet exciton expands along the direction of maximal orbital $\pi$-overlap in the crystal $a,c$ plane to form correlated triplet pairs, which subsequently decouple into free triplets along the crystal $b$-axis due to molecular sliding motion of neighbouring pentacene molecules. The fast formation of free triplets in pentacene results in the fast onset of isotropic triplet annihilation dynamics at high excitation densities, critical to applications of singlet fission that utilise triplets. Our technique is not limited to studying excitons in pentacene through optical pump-probe techniques on femtosecond timescales, but can be applied for other pump-probe schemes involving electron or X-ray sources over any experimentally accessible timescale[40]. Going forward our approach will enable direct insights into the transport of excitations in a range of condensed matter systems over a variety of timescales.

## Methods
### Pump-probe microscope setup
The design of the pump-probe microscopy setup was detailed previously[12,17]. Here, a pump pulse (560 nm, 13 fs) is focused onto the sample with the objective to produce a near-diffraction-limited local

photoexcitation. After a variable time delay, a counter-propagating widefield probe pulse (750 nm, 7 fs, ~20 μm full-width-half-maximum) is transmitted through the sample and imaged onto an emCCD camera (Rolera Thunder, QImaging). Widefield probe images in the presence and absence of the pump excitation are recorded by chopping the pump pulse at 40 Hz. The pump and probe pulses are derived from a Yb:KGW amplifier (1030 nm, 5 W, 200 kHz, 200 fs, LightConversion) via white-light-continuum generation and subsequent spectral filtering and compression with chirped mirrors (Supplementary Note 1)[12,17].

### Fabrication of the hBN-pentacene sample
hBN crystals were synthesized at 4 GPa and 1600 °C for 240 hours with Ba-BN solvent by using Belt-type high pressure apparatus. 100 nm thick hBN flakes were exfoliated from bulk crystals on a glass substrate. 100 nm on pentacene was then evaporated on the sample. Thin films of pentacene were prepared by thermal evaporation in an ultrahigh vacuum environment of $10^{-8}$ mbar at a constant evaporation rate of 0.02 nm/s. The rate was monitored using a calibrated quartz crystal microbalance and the deposition was stopped once the desired film thickness of 100 nm was obtained. The deposition was done onto 1 mm-thick quartz substrates and 0.2 mm-thick glass coverslips that were cleaned by sequential sonication in acetone and isopropyl alcohol.

## Data availability
The data that support the plots within this paper and other findings of this study are available at the University of Cambridge Repository (https://doi.org/10.17863/CAM.88392).

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

## Acknowledgements

A.A. and C.S. thank René Lachmann for stimulating discussions on optical modelling. A.A. acknowledges funding from the Gates Cambridge Trust as well as support from the Winton Programme for the Physics of Sustainability. A.V.G. acknowledges funding from the European Research Council Studentship and Trinity-Henry Barlow Scholarship. C.S. acknowledges financial support from the Royal Commission of the Exhibition of 1851. K.W. and T.T. acknowledge support from JSPS KAKENHI (Grant Numbers 19H05790, 20H00354 and 21H05233) and A3 Foresight by JSPS. We acknowledge financial support from the EPSRC via grants EP/M006360/1 and EP/W017091/1 and the Winton Program for the Physics of Sustainability. This project has received funding from the European Research Council (ERC) under the European Union's Horizon 2020 research and innovation programme (grant agreement no. 758826).

## Author contributions

A.A. conceived the project, built the optical model, performed the optical experiments, analysed and interpreted the data and wrote the manuscript. N.G. designed the sample, exfoliated the hBN and performed the AFM measurements. A.V.G. and N.S. prepared the pentacene films. A.A., A.J.S. and J.S. built the experimental setup. K.W. and T.T. provided the hBN crystals. C.S. and A.R. supervised the project. All authors discussed the results and contributed to writing the manuscript.

## Competing interests

The authors declare no competing interests.
