## [Peer Review File · Nature Communications]

Direct Observation of Ultrafast Singlet Exciton Fission in Three DimensionsREVIEWER COMMENTS

Reviewer #1 (Remarks to the Author):

The manuscript by Ashoka et al. presents interferometric pump-probe microscopy to track the quantum coherent phenomenon of singlet fission in 3D exploiting a first-principles analytic model. Their approach demonstrates that it is possible to quantify changes of the full complex refractive index and obtain information about dynamics in plane and out-of-plane with ultrafast resolution.

The study introduces several novelties in terms of: (i) experimental methodology, (ii) first-principle analysis; (iii) new physical insight on the singlet fission mechanism. Thus it represents an original and significant piece of work for the ultrafast molecular spectroscopy community and related fields, pushing the state-of-the-art literature even further. The authors are indeed actively contributing to this field even with few recent publications in this journal eg. <https://doi.org/10.1038/s41467-021-26617-w> or <https://doi.org/10.1038/s41467-022-29112-y>, however these previous studies do not overlap with the current manuscript which in my opinion provides unique, very clear results and powerful data analysis that deserve publication in the Journal of Nature Communications.

The data are well presented, and the general interpretation and conclusion are convincing, however I have few comments/ suggestions that should be addressed before considering publication, here listed.

1. The authors claim 5 nm out-of plane precision of their method, but it is not clear how this value is extracted. While pump and probe pulses are temporally well characterized, the value of the diffraction-limited pump spot is missing, to provide an estimation of in-plane spatial resolution. Also, the procedure on how to extract the motion across the different planes (eg- σ_{ac} , σ_{bc} , in Figs 2-4) should be more precisely described, maybe with a dedicated session in the SI. These details are needed for the work to be reproduced.

2. Some clarifications are needed for the modeling part: (a) is not clear the origin of eq.1> is this written in the time or frequency domain? Could the authors provide some references? (for example in ref. PRB 85, 245423 (2012) which uses similar expression for a Au thin film, there is a factor of $1/2$ instead of ϵ_0); (b) the I_{pu} expression before Eq.2 is true only if (i) E_{pu} is real amplitude (but the field is in general complex) (ii) 'n' a the numerator is the static one, but later they referred to it as n_0 .

3. Eq.3 is true only if α is dimensionless, but this seems to be not the standard expression of the Beer-Lambert equation, maybe they should make it explicit.

4. From Eq3, when Dn_0 e Dk_0 are introduced, it seems a bit confusing to follow the definition of the static value of n (n_0), and dynamic n (with no subscript). Maybe they could clarify a bit more and provide the explicit definition of Dn_0 e Dk_0 in the SI.

5. In equation 5, the authors describe the propagation of the E field across several interfaces (air-sample; sample-substrate; substrate-air). Where do they consider the substrate-air interfaces? The refractive index of $n_{\text{substrate}}$ seems not to be there for the last interface.

6. The formation of the correlated triplet pair is quantum, as stated also by the authors. Do they observe any ultrafast coherent oscillations in the dynamics? Will their analysis be able to capture that?

7. The sketch in Fig 2g and 3g describe the motion along the b-axis, but is a bit misleading: should it be representative of the pentacene on glass only, or also of pentacene of hBN?

8. For the annihilation part, the very last sentence seems a bit confusing

“The fast 200 fs motion along the b-crystal axis seen in the low-fluence regime at 250 μ J

also appears to be present in the high-density regime, suggesting that this feature

is related to the singlet fission process”. Is the 200fs assigned by the authors to the free TTA also observed at low density? At low density a time constant of 115fs is assigned to the decoupling of correlated triplets to separated triplets T...T. Where the 200 fs component appear at low density? Maybe I have missed this point.

Reviewer #2 (Remarks to the Author):

The manuscript deals with an observation of singlet exciton fission dynamics in thin pentacene films with a temporal resolution faster than the process and a spatial resolution potentially capturing late phases of the process. The authors use a pump-probe light scattering technique within a high numerical aperture microscope: the ultrashort pump laser pulse generates a singlet exciton density in the Pc sample, which expresses an index of refraction distribution, off which the probe pulse is scattered. Proper imaging provides lateral resolution as well as depth information via a short Rayleigh length. The authors claim 5 nm spatial and 15 fs temporal resolution and interpret expansion of the excitation density on two different time scales as two steps in the fission process, namely the the initial triplet pair formation and the loss of electronic correlation. Furthermore, they assign specific lattice planes and directions to these dynamics.

major comments:

The authors must introduce singlet exciton fission in pentacene along the current understanding as it is described for example in Myata et al., Chem Rev 119, 4261 (2019):

$S_0 \rightarrow S_1$, S_1 having charge transfer character in a-b plane;

$S_1 \rightarrow 1(TT)$ formation of electronic and spin coupled triplet pair (ultrafast);

$1(TT) \rightarrow 1(T...T)$ loss of electronic correlation;

$1(T...T) \rightarrow T + T$ loss of spin correlation.

Their simplified, less defined version is confusing in the introduction and prevents clear assignment of processes later in the discussion.

The description of the optical setup in section II needs to be adjusted to the readership:

the authors should condense the entire section to a simple intuitive description: "one beam (ultrashort pulse) writes a index of refraction pattern (complex) into the sample, the second beam (ultrashort pulse) is scattered thereof and transported to the camera; pattern is excitation density, delay captures spatial evolution thereof. Wavelength selects electronic transition." that's it. Physicists understand this. All (advanced text book) formulas should either go to an applied optics or scientific instrumentation journal, together with instructive executed examples with real numbers of n , k , σ , χ , λ , E .

The authors do not provide evidence for the substantial spatial (far below diffraction limit) and temporal resolution of their apparatus. A FROG trace does not resemble the real temporal resolution in the sample. Appropriate images of signal evolution, such as partially in their ms Sung2020, would be appropriate, even if in the supplementary material. A concise, quantitative assessment of their astonishing spatial resolution should be given here (section II ?)

It is not clear what the reader is supposed to learn from Figure 1d: the structure on the photo shows what? Why are there 50 molecules hardly separable in their pseudo 3 d presentation rather than for example two pairs in a useful size and the crystal axes assignable? In particular the tiny figure does not enlighten the author's most specific result being the anisotropy of exciton diffusion in the crystal lattice.

sect III A. B. C.

general remark:

I suggest that the order of discussion follows the temporal order of the process, see above: swap section A and B.

sect III A:

The manuscript does not properly distinguish between the two decoupling steps, see general comment above. And: it is not clear, what the authors mean with "motion". The experimental result is a broadening of the exciton density, observed in a specific lattice plane, due to exciton diffusion. "motion" however suggests the observation of a path, which is not possible in the described repetitive ensemble experiment. Related: what do the authors mean with "contraction, expansion". A polaron, being a local lattice response to a charge transfer?

The lack of conciseness here prevents conclusive interpretation of their interesting experimental results: The claim that "modulation of J-coupling between the triplet states of neighboring Pc molecules completes triplet separation within less than one sliding oscillation" is highly speculative. (what slides against what? entire molecules as in the cited ms?) The observable is spatial narrowing of an ensemble (exciton density). How can this be related to an individual exciton wavefunction size?

sect III B:

(related to above) Is the 25nm expansion the formation of the charge transfer exciton after $S_0 \rightarrow S_1$ excitation? How does this relate to the first principle calculations of Sharifzadeh et al., Phys Chem Lett 4, 2197 (2013), who report a few nm CT exciton size (and a lattice contraction presumably rather than an expansion)?

Figures 2,3,4 are poorly presented: what is to be learned from (a)? amplitude? sign? symmetry? Where is the 5 nm resolution in (a)? Can (b) be utilized to demonstrate spatial resolution? Why does the $\Delta T/T$ signal in 1(c) change sign upon propagation? What does the sketch in (g) mean? a spread of exciton density? Why are there two maxima along b in Fig 2? see comment above about presentation of lattice and interpretation of "expansion".

minor comments:

title: "three dimensions" are two words

the authors strive to stress the term "coherence" or "quantum coherence" very often, without explaining what they mean: In the context of ultrafast photo-excitation of molecules this could be coherent evolution of vibrational wavepackets (which it isn't here) or of electronic wavepackets (which it partially is here) or of spin coupling (which it partially is here) or simply a well determined relative/mutual phase of any two or more wavefunctions.

In the accepted understanding of exciton fission in polyacenes, simply the phase relation (coherence) between electronic states of T and T is lost first, and the spin alignment (coherence) is lost later. I suggest they concisely describe the process in this way.

pg 2 para 2: what is "part-absorptive part refractive"?

Transient absorption signals are related to one transition. The authors use the concept of joint density of states JDOS, which unnecessarily adds unspecific qualitative information and confuses the arguments. Insertion of a simple energy diagram with the observed transitions would help.

Isn't triplet fusion / annihilation in pentacene an endothermic process? (Being the reason why fission is so efficient.) How can fusion then occur on a 200 fs time scale?

In summary, while the experimental results are certainly good, the lack of stringency and clarity makes the manuscript unsuitable for publication in Nature Communications.

Reviewer #3 (Remarks to the Author):

The paper utilized a wide field ultrafast pump probe microscopic technique to study the singlet fission and exciton diffusion mechanisms in pentacene molecules deposited on various substrates. The refractive index changes extracted by fitting based on pure optical model suggested the $S_1 \rightarrow TT$ and $TT \rightarrow T \dots T$ proceed along different directions. The results are intriguing and the proposed mechanism is of fundamental importance to the singlet fission research field. However, there are a few concerns regarding the explanation of the data which should be solved before the paper can be considered for publication.

1. What are the polarization directions for pump and probe beams? I suppose they are in plane with the substrate surface? It is well known that the triplet transition for pentacene is along its long axis, that is,

“b axis” denoted in this paper. Therefore, it is highly possible that the missing of motion signal in “a,c plane” of SiO₂ and “a plane” of hBN in Fig2e is due to the polarization mismatch (which is not able to detect the signal change due to dissociated triplets) instead of a directional transport mechanism as proposed in this paper.

2. Similar to the above question, the singlet transition for pentacene is along its short axis, that is “c axis” denoted in this paper. It can explain why there is no visible motion signal in Δz_a for hBN in Fig. 3e.

3. In the microscope setup the pump is highly focused into a tiny spot. It may generate local/transient thermal effect which changes the refractive index of the sample. In fact, the authors have previously reported the thermal effect influence on the TA spectra and kinetics for pentacene on different substrates (PHYSICAL REVIEW B 84, 195411, 2011) under pump fluence of 120 $\mu\text{J}/\text{cm}^2$. The fluence used in this paper is twice higher (250 $\mu\text{J}/\text{cm}^2$), why the thermal effect is completely ignored in this study?

4. Minor typos in Supplementary materials : page 1, section 2, line 2 (in particular, Eqn. ??)

Authors' Response to Referees' Comments: NCOMMS-22-18827

We thank the three referees for carefully going through our manuscript and for their valuable comments and suggestions. In light of their comments we have made major revisions to the manuscript and supplementary materials (highlighted in blue), which we believe have made the manuscript clearer to understand and substantially stronger. We summarize below the main changes carried out. We follow it up with point by point response to the referees' comments.

1. We have extracted the confidence interval on the fits so that we can report a well-defined localisation precision rather than using the peak-to-peak noise as an estimator. We have created a dedicated section in the SI discussion on calculation of the axial and localisation precision.
2. We have added four new sections to the SI including a measurement of the pump spot size, a description of the fitting procedure, a demonstration of the high time resolution and a discussion and report of the localisation precision.
3. We have remade all of our figures to include the spatio-temporal data as well as more example images with fits to the developed optical model to demonstrate both the high time resolution and the high quality fits throughout the measurement window. Further, to aid the discussion we have added the schematic of pentacene on SiO₂ and hBN in all the figures.
4. We have substantially revised our introduction to the singlet fission process to make it more precise and have also revised our interpretation of the experimental data in light of the reviewer comments.

Point-wise responses:

Reviewer #1 (Remarks to the Author):

The manuscript by Ashoka et al. presents interferometric pump-probe microscopy to track the quantum coherent phenomenon of singlet fission in 3D exploiting a first-principles analytic model. Their approach demonstrates that it is possible to quantify changes of the full complex refractive index and obtain information about dynamics in plane and out-of-plane with ultrafast resolution.

The study introduces several novelties in terms of: (i) experimental methodology, (ii) first-principle analysis; (iii) new physical insight on the singlet fission mechanism. Thus it represents an original and significant piece of work for the ultrafast molecular spectroscopy community and related fields, pushing the state-of-the-art literature even further. The authors are indeed actively contributing to this field even with few recent publications in this journal eg. <https://doi.org/10.1038/s41467-021-26617-w> or <https://doi.org/10.1038/s41467-022-29112-y>, however these previous studies do not overlap with the current manuscript which in my opinion provides unique, very clear results and powerful data analysis that deserve publication in the Journal of Nature Communications.

We thank the reviewer for their concise summary of the manuscript and positive endorsement of the work.

The data are well presented, and the general interpretation and conclusion are convincing, however I have few comments/ suggestions that should be addressed before considering publication, here listed.

1. *The authors claim 5 nm out-of plane precision of their method, but it is not clear how this value is extracted.*

We agree that this point was not made very clear in the initial manuscript. To remedy this, we have extracted the 1σ confidence intervals on the fit extracted values for the lateral and axial localization precision. The 5-nm axial localization precision is the best case we report, and **we have included a full discussion of this in the Supplemental Material S6.**

While pump and probe pulses are temporally well characterized, the value of the diffraction-limited pump spot is missing, to provide and estimation of in-plane spatial resolution.

We thank the reviewer for pointing this out. **We have now included a measurement of the diffraction-limited pump spot by scanning fluorescent beads across the pump spot in the Supplemental Material S1, to clarify this point.**

Also, the procedure on how to extract the motion across the different planes (eg- σ_{ac} , σ_{bc} , in Figs 2-4) should be more precisely described, maybe with a dedicated session in the SI. These details are needed for the work to be reproduced.

We appreciate the reviewers point about the need for a fuller discussion on the fitting procedure and extraction of σ_{ac} , σ_{bc} etc to ensure reproducibility. **We have added dedicated section in the Supplemental Material (Section S5) on the fitting procedure and the way that these values are extracted for the different crystal planes.** Further, to aid the discussion in the main text we have added the schematic of pentacene on SiO_2 and hBN in all the figures.

2. *Some clarifications are needed for the modeling part: (a) is not clear the origin of eq.1> is this written in the time or frequency domain? Could the authors provide some references? (for example in ref. PRB 85, 245423 (2012) which uses similar expression for a Au thin film, there is a factor of 1/2 instead of ϵ_0);*

We thank the reviewer for pointing out that we had left this out. **We have now modified the introductory paragraph of the optical model in the manuscript** to ensure that the origin of Eqn. 1 from Maxwell's equations is clear:

“Before the pump pulse arrives, when the system is in the ground state (pump-off), the polarisation P measured by the probe is given by, $P = \epsilon_0\chi^{(1)}E$. Using $D = \epsilon E = \epsilon_0E + P$, the ground state dielectric function ϵ_{off} is therefore given by, $\epsilon_{\text{off}} = \epsilon_0(1 + \chi^{(1)})$. After the arrival of the pump pulse, the system is in the excited state (pump-on), the polarisation P measured by the probe is given by, $P = \epsilon_0(\chi^{(1)}E + \chi^{(3)}E_{\text{pu}}E_{\text{pu}}E)$, where E_{pu} is the pump electric field. Similarly, the excited state dielectric function ϵ_{on} is therefore given by, $\epsilon_{\text{on}} = \epsilon_0(1 + \chi^{(1)} + \chi^{(3)}E_{\text{pu}}E_{\text{pu}})$. As the probe pulse is always temporally separated from the pump, time-ordering allows us to treat the pump-on and pump-off probe signals as measures of the photoexcited and ground state dielectric functions (or complex refractive index) of the material, respectively.”

We note that the factor of a half that is referred to in the PRB as well as the other terms are a matter of convention in defining the polarization expansion and would only change the pre-factors of our model and not the underlying transport that we measure.

(b) the I_{pu} expression before Eq.2 is true only if (i) E_{pu} is real amplitude (but the field is in general complex) (ii) 'n' a the numerator is the static one, but later they referred to it as n_0 .

As the expression for I_{pu} is for the absorbed pump pulse, the effect of the pump electric field is time averaged and hence we can use the real amplitude for the electric field of the pump.

We have now clarified this in the manuscript and have corrected the n in the numerator to n_0 as the sample is always in the ground state before the arrival of the pump pulse.

3. *Eq.3 is true only if α is dimensionless, but this seems to be not the standard expression of the Beer-Lambert equation, maybe they should make it explicit.*

We thank the reviewer for pointing this out and now state this explicitly in the manuscript. We chose to use the dimensionless (thickness corrected) version of α to simplify the expressions as much as possible. **We have added a line to clarify this in the manuscript after Eqn. 3.**

4. *From Eq3, when Dn_0 e Dk_0 are introduced, it seems a bit confusing to follow the definition of the static value of n (n_0), and dynamic n (with no subscript). Maybe they could clarify a bit more and provide the explicit definition of Dn_0 e Dk_0 in the SI.*

We thank the reviewer for pointing this out. **We now state the explicit expressions in the main text to ensure that the reader can follow the definitions without having to refer to the SI. We have also corrected any ambiguous notation throughout the manuscript.**

5. *In equation 5, the authors describe the propagation of the E field across several interfaces (air-sample; sample-substrate; substrate-air). Where do they consider the substrate-air interfaces? The refractive index of $n_{\text{substrate}}$ seems not to be there for the last interface.*

The reviewer points out an important point that we did not explicitly mention in the initial version of the manuscript. As the microscope system we study is an oil-immersion objective, there is no substrate-air interface – the substrate is in contact with oil which is in contact with the objective lens. These optical elements are all designed to be refractive index matched to ensure the high numerical aperture of the objective and hence the reflections and propagation across these interfaces are not taken into consideration as the entire collection of refractive index matched interfaces is treated as a single optical layer.

We have now added a clarification on this this after Eqn. 5 in the manuscript.

6. *The formation of the correlated triplet pair is quantum, as stated also by the authors. Do they observe any ultrafast coherent oscillations in the dynamics? Will their analysis be able to capture that?*

We thank the reviewer for this intriguing proposition. In principle we should be able to see these coherent oscillations as they have been previously reported in pentacene. However, as our analysis decomposes the spectral signatures into their dn and dk components, we are at this point in time unsure where and how these oscillations will manifest. A Fourier transformation of our data is certainly possible, however as we have measured with small time steps only out to 500 fs, it is currently difficult to see any clear oscillations.

We would like to further highlight, that our technique would have to improve in signal to noise ratio (SNR) by approximately 1-2 orders of magnitude to capture such coherent oscillations. Taking a signal magnitude of about $<0.1\%$ DTT (to stay in the low fluence regime) with a 1% oscillatory amplitude would require a SNR of $<10^{-5/6}$ which is not possible with currently available two-dimensional CCD or CMOS detectors. In addition, given the rapid timescale of singlet fission in pentacene, such coherent oscillations are expected to dipphase on the < 500 fs time scale, further challenging their detection.

While detecting such oscillations is beyond the scope of our work, we believe that it should be feasible in the future to attain the required SNR.

7. *The sketch in Fig 2g and 3g describe the motion along the b-axis, but is a bit misleading: should it be representative of the pentacene on glass only, or also of pentacene of hBN?*

We have remade all four figures and added the schematic of pentacene on SiO₂ and hBN in all the figures to guide the reader through our discussion. We have removed the previous schematics of the motion from all three figures as we have found that it confused the reviewers and representing 3D motion in a figure is not a trivial task. We therefore chose to let the data describe the 3D motion instead.

8. *For the annihilation part, the very last sentence seems a bit confusing*

“The fast 200 fs motion along the b-crystal axis seen in the low-fluence regime at 250 μJ

also appears to be present in the high-density regime, suggesting that this feature

is related to the singlet fission process”. Is the 200fs assigned by the authors to the free TTA also observed at low density? At low density a time constant of 115fs is assigned to the decoupling of correlated triplets to separated triplets T...T. Where the 200 fs component appear at low density? Maybe I have missed this point.

This is a typo in the initial manuscript which we apologies for. We meant to write sub-200 fs component (which is the 115 fs component seen in the low-density regime at 740 nm and 70 fs component seen at 670 nm). **We have rectified this in the new version of the manuscript.**

Reviewer #2 (Remarks to the Author):

The manuscript deals with an observation of singlet exciton fission dynamics in thin pentacene films with a temporal resolution faster than the process and a spatial resolution potentially capturing late phases of the process. The authors use a pump-probe light scattering technique within a high numerical aperture microscope: the ultrashort pump laser pulse generates a singlet exciton density in the Pc sample, which expresses an index of refraction distribution, off which the probe pulse is scattered. Proper imaging provides lateral resolution as well as depth information via a short Rayleigh length.

We thank the reviewer for their summary of the manuscript. We would like to add that the depth information that allows for three-dimensional tracking arises from interferometric contrast rather than just the short Rayleigh length.

The authors claim 5 nm spatial and 15 fs temporal resolution and interpret expansion of the excitation density on two different time scales as two steps in the fission process, namely the the initial triplet pair formation and the loss of electronic correlation. Furthermore, they assign specific lattice planes and directions to these dynamics.

major comments:

The authors must introduce singlet exciton fission in pentacene along the current understanding as it is described for example in Myata et al., Chem Rev 119, 4261 (2019):

S0 -> S1, S1 having charge transfer character in a-b plane;

S1 -> I(TT) formation of electronic and spin coupled triplet pair (ultrafast);

I(TT) -> I(T...T) loss of electronic correlation;

I(T...T) -> T + T loss of spin correlation.

Their simplified, less defined version is confusing in the introduction and prevents clear assignment of processes later in the discussion.

We thank the reviewer for pointing this out. We have now read the cited paper and have substantially revised all of our definitions and explanations surrounding the singlet fission process. **Specifically we now introduce singlet fission in the Introduction section as,**

Singlet exciton fission is a widely studied example of such a process that has gained relevance in the fields of photovoltaics and quantum computing. Here a photo generated singlet exciton converts to an electronically and spin entangled pair of triplets at nearly half the singlet energy on ultrafast timescales. The correlated triplet pair then separates into individual triplet excitons through the loss of electronic and spin coherence [8–11].

In the results section we describe the process as,

The process of ultrafast singlet fission is considered to occur in three steps. First, the photogenerated singlet S1 converts to an electronically and spin entangled triplet pair state T T . Second, through the loss of electronic correlation the TT state converts to a spatially separated triplet pair T...T. Finally, through the loss of spin correlation, the spatially separated spin entangled triplet pair T...T separates into two uncorrelated triplets T + T [8]. As spin coherence is typically lost on longer timescales than the electronic coherence in polyacenes and as we probe the electronic states through their optical transitions, we solely focus on the

loss of electronic correlation from TT to T...T and make no comment on the spin correlation [8].

Finally, we have amended the title of Section III A to properly reflect the process we are studying and avoid the spin vs electronic decoherence ambiguity.

The description of the optical setup in section II needs to be adjusted to the readership:

the authors should condense the entire section to a simple intuitive description: "one beam (ultrashort pulse) writes a index of refraction pattern (complex) into the sample, the second beam (ultrashort pulse) is scattered thereof and transported to the camera; pattern is excitation density, delay captures spatial evolution thereof. Wavelength selects electronic transition." that's it. Physicists understand this. All (advanced text book) formulas should either go to an applied optics or scientific instrumentation journal, together with instructive executed examples with real numbers of n , k , σ , χ , λ , E .

We thank the reviewer for their suggestion. **In light of this we have modified the introductory part of Section II to ensure that the condensed version of the description of the optical setup is available to the reader as follows:**

The effect of the pump pulse is to photoinduce a three-dimensional spatially varying, local complex refractive index change, $\Delta\tilde{n} = \Delta n + i\Delta k$ (Fig 1a). This index change weakly perturbs the time-delayed plane-wave probe pulse incident on the sample, leading to local changes in its phase and amplitude [18]. The large background unperturbed probe field interferes with this perturbed probe field to form a spatial interference pattern along the propagation direction. The objective and imaging lens then image this spatial interference pattern combined with the attenuation of the probe on a camera.

We have retained the mathematical description of the optical model as we believe it was appreciated by the other two reviewers. Further, as this is a non-subject specific journal not aimed specifically at physicists, we believe that this allows for the general reader to follow the paper more easily. Based on other reviewer comments, we have also given more details on the derivation of certain formulas to ensure ease of readability.

The authors do not provide evidence for the substantial spatial (far below diffraction limit) and temporal resolution of their apparatus.

The spatial precision which we report is not the optical resolution of the setup, which is fundamentally set by the diffraction limit. Several papers in the fields of ultrafast microscopy and single molecule localization microscopy have discussed the differences (1-4). **We now clarified this in the Supplemental Material S6.** Furthermore, we have now extracted the 1σ confidence intervals on the fit extracted values for the lateral and axial localization precision. The 5-nm axial localization precision is the best case we report, and we have included a full discussion of this in the Supplemental Material S6.

A FROG trace does not resemble the real temporal resolution in the sample. Appropriate images of signal evolution, such as partially in their ms Sung2020, would be appropriate, even if in the supplementary material.

All of the optical elements used in the FROG measurement are carefully selected to ensure that the pulses experience the same path as the experiment (identical microscope objective, waveplate etc). We therefore believe that the FROG trace serves as a good proxy for the time resolution in the sample plane in the microscope. **We have amended Supplemental Material**

S1 to ensure that this point is communicated clearly. At the reviewer's request we have also **added a section to Supplemental Material S4** with images of signal evolution of the first 60 fs as in the Sung 2020 paper. We have **also remade Figures 2-4** and added in panel (a) of Figures 2,3 and 4 a spatio-temporal map of the signal along with the average kinetic to demonstrate the presence of ultrafast dynamics and our time resolution. We have further added 2 more images and fits in panel (b) of Fig. 2-4 to also showcase the signal evolution.

A concise, quantitative assessment of their astonishing spatial resolution should be given here (section II ?)

We stress that the spatial precision which we report is not the resolution of the setup which is still fundamentally set by the diffraction limit. We believe that this was unclear in the first manuscript and have now added a section Supplemental Material S6 to discuss this fully where we have extracted the 1σ confidence intervals on the fit extracted values for the lateral and axial localization precision. **The 5-nm axial localization precision is the best case we report, and we have included a full discussion of this in the Supplemental Material S6.**

It is not clear what the reader is supposed to learn from Figure 1d: the structure on the photo shows what? Why are there 50 molecules hardly separable in their pseudo 3 d presentation rather than for example two pairs in a useful size and the crystal axes assignable? In particular the tiny figure does not enlighten the author's most specific result being the anisotropy of exciton diffusion in the crystal lattice.

We thank the reviewer for their criticism of our figures. **We have now remade all of our Figures and in Figure 1 we have now reduced the number of molecules to bring out the anisotropy in the crystal lattice as suggested.** Further, to aid the discussion section we have added the schematic of pentacene on SiO₂ and hBN in all the figures and removed the confusing schematic of the exciton density motion.

sect III A. B. C.

general remark:

I suggest that the order of discussion follows the temporal order of the process, see above: swap section A and B.

We thank the reviewer for their suggestion. We initially considered presenting the data like this, however this would require the results and discussion section to begin with the photobleach at 670 nm which contains the convoluted spectral information of both the singlet and triplet excitons. Based on our analysis, it is not possible to assign the measured transport at 670 nm without first having the complementary information from the 740 nm band of the triplet transport. Hence, we chose to present the data in the order that is easier to follow rather than following the temporal order of the process. We hope the reviewer sees why this choice makes the manuscript easier to follow without forcing the reader to move back and forth between datasets too often. **We have also added the following sentence to the manuscript to ensure the reader understands this decision.**

We begin by studying the uncongested photoinduced absorption band at 740 nm to establish the spatial dynamics of the triplet excitons and use this information to study the more complicated dynamics of the ground state bleach band at 670 nm.

sect III A:

The manuscript does not properly distinguish between the two decoupling steps, see general comment above.

We thank the reviewer for this comment. We have now included an explicit mention of the decoupling processes that we focus on in this paper (the loss of electronic correlation). As we focus on the ultrafast timescales and optical transitions, the loss of electronic correlation is the main process we are sensitive to. **We have therefore included the following line in the manuscript:**

As spin coherence is typically lost on longer timescales than the electronic coherence in polyacenes and as we probe the electronic states through their optical transitions, we solely focus on the loss of electronic correlation from TT to T...T and make no comment on the spin correlation [8].

And: it is not clear, what the authors mean with "motion". The experimental result is a broadening of the exciton density, observed in a specific lattice plane, due to exciton diffusion. "motion" however suggests the observation of a path, which is not possible in the described repetitive ensemble experiment.

We thank the reviewer for pointing this out. We agree with the reviewer that this is confusing language and have therefore strived to **remove the word motion from the manuscript, replacing it with 'transport' where appropriate.** These changes are annotated throughout the manuscript and we hope the reviewer finds it suitably clear now.

Related: what do the authors mean with "contraction, expansion". A polaron, being a local lattice response to a charge transfer?

We agree that these phrases are difficult to interpret and have strived to remove them where possible in the manuscript and instead mention the change in the underlying exciton density that we are actually probing. We note that as the entire model and optical transitions used track the exciton density, all of the previous references to contraction or expansion necessarily refer to changes in the exciton density. Physical assignment of the origin of this change from a polaronic distortion or other processes is not straightforward from spectroscopy alone. We have therefore opted to adopt a more high-level description to not falsely imply physical insight. **More theoretical support and experiments are needed to clarify this and we have added the following line in the manuscript to reflect this.**

Further theoretical investigations of the response of the excitonic wavefunction to phonon modes along this axis in the vibronic and transient delocalisation framework are called for [33–35].

The lack of conciseness here prevents conclusive interpretation of their interesting experimental results: The claim that "modulation of J-coupling between the triplet states of neighboring Pc molecules completes triplet separation within less than one sliding oscillation" is highly speculative. (what slides against what? entire molecules as in the cited ms?) The observable is spatial narrowing of an ensemble (exciton density). How can this be related to an individual exciton wavefunction size?

We agree with the reviewer and apologize for our lack of conciseness. Unfortunately, the nature of our measurement makes a clear mechanistic insight challenging from this measurement

alone, even though the phenomenology of what we observe is clear. The sliding motion we refer to is due to the same lattice modes as in the cited paper. **We have rewritten this section as follows to reflect the key insights we can draw and refrain from claiming more to avoid speculation.**

Hence this process must be related to the loss of electronic correlation in the triplet pair. Recent reports suggest that a 1 THz (1 ps period) lattice vibration in pentacene crystals associated with sliding motion of neighbouring pentacene molecules along the crystal b-axis modulates the π -overlap and therefore the J-coupling between adjacent pentacene molecules, resulting in triplet pair separation from $TT \rightarrow T...T$ [32]. Our measured 6 nm reduction in the spatial triplet exciton density along the same crystal b-axis over 500 fs could be related to a lattice distortion arising from this 1 THz mode which changes the local excitonic density along this axis. A full 1 ps oscillation period of the 1 THz sliding mode is not needed to decouple the TT state to the $T...T$ state, and as we show below, free triplets are formed within 200 fs of photoexcitation. We additionally note that the triplet exciton in pentacene is known to be polarised along the b-axis which further suggests a strong change in polarizability due to this mode [33]. Further theoretical investigations of the response of the excitonic wavefunction to phonon modes along this axis in the vibronic and transient delocalisation framework are called for [33–35].

We hope that the reviewer finds that this section appropriately rewritten while retaining the novelty of our observations without speculating beyond the reasonable.

sect III B:

(related to above) Is the 25nm expansion the formation of the charge transfer exciton after $S_0 \rightarrow S_1$ excitation? How does this relate to the first principle calculations of Sharifzadeh et al., Phys Chem Lett 4, 2197 (2013), who report a few nm CT exciton size (and a lattice contraction presumably rather than an expansion)?

We believe that the 25 nm expansion could be related to the formation and subsequent transport of the charge transfer state and **we now mention this in the manuscript**. We believe that the timescales for this transport (70 fs) is too fast to allow for a lattice reorganization around the exciton, however we cannot rule this out but just choose not to comment on it at this stage.

Figures 2,3,4 are poorly presented: what is to be learned from (a)? amplitude? sign? symmetry?

We thank the reviewer for their criticism. **We have now remade all of these figures**. In particular, we have changed (a) to show the temporal evolution of the radially averaged spatial signal along with an averaged kinetic to demonstrate the time resolution. We have further added in 3 example images and fits in panel (b) to demonstrate that the model fits equally well at all timepoints. Also, to aid the discussion we have added the schematic of pentacene on SiO_2 and hBN in (c) in all the figures.

Where is the 5 nm resolution in (a)?

The 5 nm axial precision is merely the best-case scenario with our current signal to noise at 740 nm where the imaging plane chosen does not have substantial aberrations that reduce the confidence of the fit. It is clear from the data that the axial precision is worse at 670 nm as we're on the edge of the region where our remote focusing technique works well and this is also discussed in Supplemental Material S7. **We have also added the following line to the manuscript to ensure this point is communicated.**

We note that the high localisation precision is significantly diminished on this spectral band as the required interferometric contrast can only be gained through substantially defocussing ($|\Delta z| > 500$ nm at 670 nm compared to $|\Delta z| < 50$ nm at 740 nm), where the remote focussing model begins to fail (see Supplemental Material S X).

Can (b) be utilized to demonstrate spatial resolution?

The spatial resolution of our setup is detailed in Supplemental Material S1 where we use a fluorescent bead to map out the excitation profile of the pump pulse. **We now discuss a few different methods to evaluate the spatial localization precision in Supplemental Material S6.**

Why does the $\Delta T/T$ signal in 1(c) change sign upon propagation?

Fig 1c) is demonstrating the interferometric contrast available in different imaging planes in the sample rather than upon propagation. The signal changes sign due to the presence of a phase contrast imprinted on the propagating probe by the real refractive index. The same signal inversion in different imaging planes is seen in state-of-the-art interferometric contrast microscopy. This is why we have retained the mathematical expressions in Section II which are needed to fully describe the origin of this interferometric contrast.

What does the sketch in (g) mean? a spread of exciton density? Why are there two maxima along b in Fig 2? see comment above about presentation of lattice and interpretation of "expansion".

We have remade all four figures and added the schematic of pentacene on SiO₂ and hBN in all the figures. We have removed the schematic of the motion from all three figures as we have found that it confused the reviewers and representing 3D motion in a figure is not a trivial task. We therefore chose to let the data describe the 3D motion instead.

minor comments:

title: "three dimensions" are two words

We thank the reviewer for pointing this out and have now corrected this.

the authors strive to stress the term "coherence" or "quantum coherence" very often, without explaining what they mean: In the context of ultrafast photo-excitation of molecules this could be coherent evolution of vibrational wavepackets (which it isn't here) or of electronic wavepackets (which it partially is here) or of spin coupling (which it partially is here) or simply a well determined relative/mutual phase of any two or more wavefunctions. In the accepted understanding of exciton fission in polyacenes, simply the phase relation (coherence) between electronic states of T and T is lost first, and the spin alignment (coherence) is lost later. I suggest they concisely describe the process in this way.

We thank the reviewer for this comment and completely sympathize with the overuse/misuse of the phrase 'coherence'. In order to avoid potential misuse, we have **removed this phase from the manuscript** and have only used it when specifically distinguishing between the loss of spin and electronic coherence as suggested by the reviewer.

pg 2 para 2: what is "part-absorptive part refractive"?

We had intended this term to aid the intuition of the reader when thinking about the complex three-dimensional refractive index perturbation (with a finite real/refractive part and finite complex/absorptive part). **We see that is may be confusing and have therefore removed it from the manuscript.**

Transient absorption signals are related to one transition. The authors use the concept of joint density of states JDOS, which unnecessarily adds unspecific qualitative information and confuses the arguments. Insertion of a simple energy diagram with the observed transitions would help.

We respectfully disagree with the reviewer on this point. One of the novelties of our work (appreciated by Reviewer 1) is the ability to extract the transient complex refractive indices which allows access to the joint-density-of-states. We have recently published a paper on why this is important in optical spectroscopy (<https://www.nature.com/articles/s41467-022-29112-y>). However, we admit that this is not typical for spectroscopy papers and **now include the following line on why the molecular energy state picture need to be modified in extended systems.**

We describe the transitions in these systems through their JDOS rather than as single energetic transitions as in extended thin film systems, the molecules are not isolated and the density of states cannot be treated as one-dimensional.

Isn't triplet fusion / annihilation in pentacene an endothermic process? (Being the reason why fission is so efficient.) How can fusion then occur on a 200 fs time scale?

We agree that triplet fusion cannot occur on a 200 fs timescale. Triplet annihilation however has many photophysical pathways only one of which is back to the original singlet, i.e., triplet fusion. For example annihilation to higher lying triplet states is certainly possible as discussed here (<https://onlinelibrary.wiley.com/doi/10.1002/adma.201302427>). **To ensure this is clear we have added the following line to the manuscript.**

This fast triplet annihilation process is must be to higher lying triplet states and not back to the singlet manifold as the latter process is endothermic and would be expected to occur on longer timescales.

In summary, while the experimental results are certainly good, the lack of stringency and clarity makes the manuscript unsuitable for publication in Nature Communications.

We hope that given the major revisions of the main text and SI we have conducted based on the reviewer comments, the paper suitably improved in its stringency and clarity and is suitable for publication.

References

1. J. Sung, C. Schnedermann, L. Ni, A. Sadhanala, R. Y. S. Chen, C. Cho, L. Priest, J. M. Lim, H.-K. Kim, B. Monserrat, P. Kukura, A. Rao, Long-range ballistic propagation of carriers in methylammonium lead iodide perovskite thin films. *Nat. Phys.* **16**, 171–176 (2020).
2. M. Lelek, M. T. Gyparaki, G. Beliu, F. Schueder, J. Griffié, S. Manley, R. Jungmann, M. Sauer, M. Lakadamyali, C. Zimmer, Single-molecule localization microscopy. *Nat. Rev. Methods Prim.* **1** (2021), doi:10.1038/s43586-021-00038-x.

3. C. Schnedermann, J. Sung, R. Pandya, S. D. Verma, R. Y. S. Chen, N. Gauriot, H. M. Bretscher, P. Kukura, A. Rao, Ultrafast Tracking of Exciton and Charge Carrier Transport in Optoelectronic Materials on the Nanometer Scale. *J. Phys. Chem. Lett.* **10**, 6727–6733 (2019).
4. M. Delor, H. L. Weaver, Q. Q. Yu, N. S. Ginsberg, Imaging material functionality through three-dimensional nanoscale tracking of energy flow. *Nat. Mater.* **19**, 56–62 (2020).

Reviewer #3 (Remarks to the Author):

The paper utilized a wide field ultrafast pump probe microscopic technique to study the singlet fission and exciton diffusion mechanisms in pentacene molecules deposited on various substrates. The refractive index changes extracted by fitting based on pure optical model suggested the SI->TT and TT->T...T proceed along different directions. The results are intriguing and the proposed mechanism is of fundamental importance to the singlet fission research field. However, there are a few concerns regarding the explanation of the data which should be solved before the paper can be considered for publication.

We thank the reviewer for their concise summary of the manuscript and positive endorsement of the work.

1. What are the polarization directions for pump and probe beams? I suppose they are in plane with the substrate surface? It is well known that the triplet transition for pentacene is along its long axis, that is, "b axis" denoted in this paper. Therefore, it is highly possible that the missing of motion signal in "a,c plane" of SiO₂ and "a plane" of hBN in Fig2e is due to the polarization mismatch (which is not able to detect the signal change due to dissociated triplets) instead of a directional transport mechanism as proposed in this paper.

We thank the reviewer for this very interesting point. **We have added a line in the manuscript and SI about the beam polarizations and how they are optimized for the maximum signal for clarity (they are indeed in the substrate plane).** Regarding the possibility that the missing motion in 2e arises due to polarization selectivity, we note that the direction of motion along a particular crystal axis has nothing to do with the polarization of the underlying excitation. For example, even though the triplet is polarized along the b-axis, the motion is not necessarily along the b-axis and can certainly move in the a,c plane, which would be picked up by our measurements which are sensitive to the local triplet exciton density. However, it is true that we would miss motion of the orthogonally polarized singlet exciton due to a polarization mismatch, however, the photoinduced absorption at 740 nm in pentacene is well established to correspond exclusively to the triplet exciton density, and hence we are already spectrally selecting the triplet excitonic population. This is why all of our observations in this spectral region correspond to transport exclusively of the triplet exciton density.

However, thanks to this point we have realized that 1 THz lattice mode along the b-axis that we refer to would naturally perturb the triplet exciton much more than the singlet exciton due to this underlying polarization preference, supporting our hypothesis that the change in exciton density we see is due to the 1 THz mode. **We have added the following line to the manuscript to reflect this.**

We additionally note that the triplet exciton in pentacene is known to be polarised along the b-axis which further suggests a strong change in polarizability due to this mode [33].

2. *Similar to the above question, the singlet transition for pentacene is along its short axis, that is “c axis” denoted in this paper. It can explain why there is no visible motion signal in Δz_a for hBN in Fig. 3e.*

Similarly, for the singlet transition, even though it is polarized along the c axis, there is no a-priori reason to expect insensitivity to or a lack of motion along the a-axis. Linear polarized dipoles are allowed to move isotopically in space unless the underlying hopping interaction is dipolar where the coupling is maximized only along certain directions. We do not consider that dipolar hopping is relevant on the fs timescales we investigate but this is certainly an important point to consider for work focusing on longer timescales. We stress that if an excitonic density polarized along the a-axis were to expand along the b-axis, our optical microscopy technique would be able to resolve this motion as we image the excitonic density in real space.

Further, we note that the 670 nm transition is the ground state bleach of the system and therefore contains contributions from both the singlet and triplet excitons as we mention in the manuscript. Hence only once we deconvolve the triplet contribution do we comment on any excitonic motion. **We have now added the following line in the manuscript to make this clearer.**

We begin by studying the uncongested photoinduced absorption band at 740 nm to establish the spatial dynamics of the triplet excitons and use this information to study the more complicated dynamics of the ground state bleach band at 670 nm.

3. *In the microscope setup the pump is highly focused into a tiny spot. It may generate local/transient thermal effect which changes the refractive index of the sample. In fact, the authors have previously reported the thermal effect influence on the TA spectra and kinetics for pentacene on different substrates (PHYSICAL REVIEW B 84, 195411, 2011) under pump fluence of 120 uJ/cm². The fluence used in this paper is twice higher (250 uJ/cm²), why the thermal effect is completely ignored in this study?*

We thank the reviewer for their comment. Based on the PRB paper cited, the thermal heating effects are linear in pump fluence, and we would therefore expect the same effects mentioned there in our measurements. The two spectral features we study here at 670 nm and 740 nm have both been established to arise from the underlying photophysics of pentacene rather than thermal artifacts as per the cited paper. Further, the motion we observe cannot be ascribed to heat transport as thermal transport typically operates on nanosecond diffusive timescales rather than the sub 2ps dynamics we study here. Together these considerations rule out thermal artifacts as the origins of our signals.

4. *Minor typos in Supplementary materials : page 1, section 2, line 2 (in particular, Eqn. ??)*

We thank the reviewer for pointing this out, we have now rectified this error.

REVIEWERS' COMMENTS

Reviewer #1 (Remarks to the Author):

I'm satisfied by the replies.

No further comments from my side.

I think the manuscript did improve and it's now suitable for publication.

Reviewer #2 (Remarks to the Author):

The reviewer thanks the authors to take his criticism and comments seriously.

The major concerns around fission model and spatial resolution are satisfactorily dealt with by restructuring and rewording and additional clarifications. Good.

Figures are much more informative and clear. Good.

Interpretation of results is appropriately close to the facts (and current understanding) and less speculative, explicitly includes limitations of analysis. Good.

To the opinion of the reviewer, the entire manuscript now reads much more concise and clear.

The reviewer appreciates the author's additional explanations, where they found entry in the ms, and also where they convinced and taught him. Thanks.

I still find the detailed mathematical derivation of the optical model inappropriate for a cross-disciplinary journal, but I will leave this issue to the editor.

In conclusion I revise my earlier conclusion and recommend the manuscript for publication.

Reviewer #3 (Remarks to the Author):

The authors answered my questions correctly. This manuscript provides clear results and data analysis. I believe that the subject of the paper is of interest. I would like to suggest that this paper to be published in Nature Communications.

Authors' Response to Referees' Comments: NCOMMS-22-18827A

We thank the three referees for carefully going through our manuscript and for their valuable comments and suggestions.

Point-wise responses:

Reviewer #1 (Remarks to the Author):

I-m satisfied by the replies.

No further comments from my side.

I think the manuscript did improve and it's now suitable for publication.

We thank the reviewer for their positive endorsement for publication.

Reviewer #2 (Remarks to the Author):

The reviewer thanks the authors to take his criticism and comments seriously.

The major concerns around fission model and spatial resolution are satisfactorily dealt with by restructuring and rewording and additional clarifications. Good.

Figures are much more informative and clear. Good.

Interpretation of results is appropriately close to the facts (and current understanding) and less speculative, explicitly includes limitations of analysis. Good.

To the opinion of the reviewer, the entire manuscript now reads much more concise and clear.

The reviewer appreciates the author's additional explanations, where they found entry in the ms, and also where they convinced and taught him. Thanks.

I still find the detailed mathematical derivation of the optical model inappropriate for a cross-disciplinary journal, but I will leave this issue to the editor.

In conclusion I revise my earlier conclusion and recommend the manuscript for publication.

We thank the reviewer for their positive endorsement for publication.

Reviewer #3 (Remarks to the Author):

The authors answered my questions correctly. This manuscript provides clear results and data analysis. I believe that the subject of the paper is of interest. I would like to suggest that this paper to be published in Nature Communications.

We thank the reviewer for their positive endorsement for publication.